# What Representational Similarity Measures Imply about Decodable Information

**Sarah E. Harvey**
Center for Computational Neuroscience
Flatiron Institute, New York, NY
`sharvey@flatironinstitute.org`

**David Lipshutz**
Center for Computational Neuroscience
Flatiron Institute, New York, NY
`dlipshutz@flatironinstitute.org`

**Alex H. Williams**
Center for Neural Science, Center for Computational Neuroscience
New York University, Flatiron Institute, New York, NY
`alex.h.williams@nyu.edu`

**Editors:** Marco Fumero, Clementine Domine, Zorah Lähner, Donato Crisostomi, Luca Moschella, Kimberly Stachenfeld

## Abstract

Neural responses encode information that is useful for a variety of downstream tasks. A common approach to understand these systems is to build regression models or "decoders" that reconstruct features of the stimulus from neural responses. Popular neural network similarity measures like centered kernel alignment (CKA), canonical correlation analysis (CCA), and Procrustes shape distance, do not explicitly leverage this perspective and instead highlight geometric invariances to orthogonal or affine transformations when comparing representations. Here, we show that many of these measures can, in fact, be equivalently motivated from a decoding perspective. Specifically, measures like CKA and CCA quantify the average alignment between optimal linear readouts across a distribution of decoding tasks. We also show that the Procrustes shape distance upper bounds the distance between optimal linear readouts and that the converse holds for representations with low participation ratio. Overall, our work demonstrates a tight link between the geometry of neural representations and the ability to linearly decode information. This perspective suggests new ways of measuring similarity between neural systems and also provides novel, unifying interpretations of existing measures.

## 1 Introduction

The computational neuroscience and machine learning communities have developed a multitude of methods to quantify similarity in population-level activity patterns across neural systems. Indeed, a recent review by Klabunde et al. [20] catalogues over thirty approaches to quantifying representational similarity. Many of these measures quantify similarity in the *shape* or *representational geometry* of point clouds. For example, recent papers (e.g. [36, 9]) leverage the Procrustes distance and other concepts from *shape theory*—an established body of work that formalizes the notion of shape and ways to measure distance between shapes [18, 10]. Other measures of representational similarity are not quite this explicit but still emphasize desired *geometric invariance* properties. For example, work by Kornblith et al. [22] popularized centered kernel alignment (CKA) by emphasizing its invariance to isotropic scaling, translation, and orthogonal transformations. These are precisely the

Proceedings of the II edition of the Workshop on Unifying Representations in Neural Models (UniReps 2024).

same invariances specified by classical shape theory [18, 10]. Earlier work by Raghu et al. [29] argued for a more flexible invariance to affine transformations, and they used canonical correlations analysis (CCA) for this purpose. Contemporary work in neuroscience is replete with similar geometric reasoning and quantitative frameworks [25].

Here we ask: What does the similarity between neural representations, potentially in terms of shape or geometry, imply about the similarity between *functions* performed by those neural systems? Potentially, not very much. Maheswaranathan et al. [27] showed that recurrent neural networks with different architectures performed the same task with similar dynamical algorithms, but with different representational geometry. More recently, Lampinen et al. [26] showed that representational geometry can be biased by other nuisance variables, such as the order in which multiple tasks are learned. Similar limitations of representational geometry measures are discussed in [11, 8].

On the other hand, several basic observations suggest that geometry and function ought to be related. Consider, for example, the common practice of using linear models to decode task-relevant variables from neural population activity [1, 24]. The premise behind these analyses is that anything linearly decodable from system $A$ is readily accessible to layers or brain regions immediately downstream of $A$. Therefore, information that is linearly decodable from $A$ may relate to functional role played by $A$ within the overall system [4]. Notably, the invariances of typical representational similarity measures— translations, isotropic scalings, and orthogonal transformations—closely coincide with the set of transformations that leave decoding accuracy unchanged. For example, translations and rotations of neural population activity will not impact the accuracy of a linear decoder with an intercept term and an $L_2$ penalty on the weights. Thus, if we accept the premise that decoding accuracy is a proxy—even, perhaps, an imperfect proxy—for neural system function, then representational geometry may be a reasonable framework to capture something about this function [23].

We provide a unifying theoretical framework that connects several existing methods of measuring representational similarity in terms of linear decoding statistics. Specifically, we show that popular methods such as centered kernel alignment (CKA) [22] and similarity based on canonical correlations analysis (CCA) [29] can be interpreted as alignment scores between optimal linear decoders with particular weight regularizations. Additionally, we study how the *shape* of neural representations is related to decoding by deriving bounds that relate the average decoding distance and the Procrustes distance. We find that the Procrustes distance is a more strict notion of geometric dissimilarity than the expected distance between optimal decoder readouts, in that a small value of the Procrustes distance implies a small expected distance between optimal decoder readouts but the converse is not necessarily true. This formalizes recent observations by Cloos et al. [5], who found that high Procrustes similarity implied high CKA similarity in practical settings.

## 2 Theoretical Framework for Linear Decoding

We use $\boldsymbol{X} \in \mathbb{R}^{M \times N_X}$ and $\boldsymbol{Y} \in \mathbb{R}^{M \times N_Y}$ to denote matrices holding responses to $M$ stimulus conditions measured across neural populations consisting of $N_X$ and $N_Y$ neurons, respectively. Throughout this paper we will assume that the neural responses have been preprocessed so that the columns of $\boldsymbol{X}$ and $\boldsymbol{Y}$ have zero mean.

### 2.1 Linear decoding from neural population responses

We first consider the problem of predicting (or "decoding") a target vector $\boldsymbol{z} \in \mathbb{R}^M$ from neural population responses by a linear function $\boldsymbol{X} \mapsto \boldsymbol{X}\boldsymbol{w}$ where $\boldsymbol{w} \in \mathbb{R}^{N_X}$. One can view this as a simplified neural circuit model where each element of $\hat{\boldsymbol{z}} = \boldsymbol{X}\boldsymbol{w} + \boldsymbol{\epsilon}$ is the firing rate readout unit the $M$ conditions. Here, we interpret $\boldsymbol{w}$ as a vector of synaptic weights and the random vector $\boldsymbol{\epsilon}$ represents potential noise corruptions; for example, $\boldsymbol{\epsilon}_i \sim \mathcal{N}(0, \sigma^2)$ where $\sigma^2 > 0$ specifies the scale of noise. This neural circuit interpretation of linear decoding is fairly standard within the literature [13, 17].

Within this setting, we formalize the problem of decoding $\boldsymbol{z}$ from the population activity $\boldsymbol{X}$ through the following class of optimization problems:

$$\underset{\boldsymbol{w}}{\text{maximize}} \quad \frac{1}{M}\boldsymbol{z}^\top \boldsymbol{X}\boldsymbol{w} - \frac{1}{2}\boldsymbol{w}^\top \boldsymbol{G}(\boldsymbol{X})\boldsymbol{w} \tag{1}$$

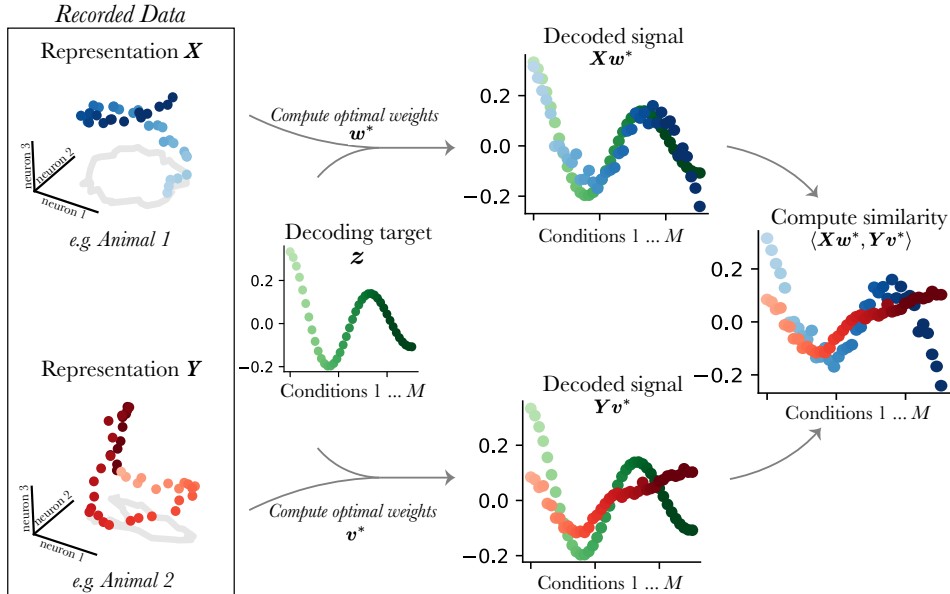

*Figure 1:* Schematic of the proposed framework for comparing representations $\boldsymbol{X}$ and $\boldsymbol{Y}$ (each dot represents mean neural responses to one of $M$ conditions) in terms of a decoding target $\boldsymbol{z}$. First, optimal linear decoding weights $\boldsymbol{w}^*$ and $\boldsymbol{v}^*$ are computed. Then the similarity between the two systems is measured in terms of the *decoding similarity*: $\langle \boldsymbol{X}\boldsymbol{w}^*, \boldsymbol{Y}\boldsymbol{v}^* \rangle$.

where $\boldsymbol{G}(\cdot)$ is a function mapping $\mathbb{R}^{M \times N}$ onto symmetric positive definite $N \times N$ matrices. The objective is concave, and equating the gradient to zero yields a closed form solution:

$$\boldsymbol{w}^* = \tfrac{1}{M} \boldsymbol{G}(\boldsymbol{X})^{-1} \boldsymbol{X}^\top \boldsymbol{z} \tag{2}$$

To appreciate the relevance of eq. (1), note that if $\boldsymbol{G}(\boldsymbol{X}) = \boldsymbol{C}_X + \lambda \boldsymbol{I}$, where $\boldsymbol{C}_X := \tfrac{1}{M} \boldsymbol{X}^\top \boldsymbol{X}$ is the empirical covariance of $\boldsymbol{X}$, then the solution $\boldsymbol{w}^*$ coincides with the optimum of the familiar ridge regression problem:

$$\underset{\boldsymbol{w}}{\text{minimize}} \quad \tfrac{1}{M} \|\boldsymbol{z} - \boldsymbol{X}\boldsymbol{w}\|_2^2 + \lambda \|\boldsymbol{w}\|_2^2 \tag{3}$$

with solution $\boldsymbol{w}^* = \tfrac{1}{M} (\boldsymbol{C}_X + \lambda \boldsymbol{I})^{-1} \boldsymbol{X}^\top \boldsymbol{z}$, where $\lambda > 0$ specifies the strength of regularization on the coefficients. More generally, we can interpret eq. (1) as maximizing the inner product between the target, $\boldsymbol{z}$, and the linear readout, $\boldsymbol{X}\boldsymbol{w}$, plus a penalty term on the scale of $\boldsymbol{w}$ as quantified by the function $\boldsymbol{w} \mapsto \sqrt{\boldsymbol{w}^\top \boldsymbol{G}(\boldsymbol{X})\boldsymbol{w}}$, which is a norm on $\mathbb{R}^N$.

Equation (1) is a useful generalization of eq. (3) because there are situations where the scale of the decoded signal matters. For instance, when the readout unit is corrupted with noise, $\hat{\boldsymbol{z}} = \boldsymbol{X}\boldsymbol{w} + \boldsymbol{\epsilon}$ where $\epsilon_i \sim \mathcal{N}(0, \sigma^2)$, we can interpret eq. (1) as maximizing the signal-to-noise ratio subject to a cost on the magnitude of the weights $\boldsymbol{w}$.

In most cases, we will consider choices of $\boldsymbol{G}(\cdot)$ that can be written as:

$$\boldsymbol{G}(\boldsymbol{X}) = a\boldsymbol{C}_X + b\boldsymbol{I} \tag{4}$$

for some $a, b \geq 0$. Under this choice, the penalty on $\boldsymbol{w}$ equals the sum of two terms:

$$\boldsymbol{w}^\top \boldsymbol{G}(\boldsymbol{X})\boldsymbol{w} = \frac{a}{M} \|\boldsymbol{X}\boldsymbol{w}\|_2^2 + b\|\boldsymbol{w}\|_2^2. \tag{5}$$

The first term, scaled by $a$, penalizes the activation level of the readout unit, which can be viewed as an energetic constraint. The second term, scaled by $b$, penalizes the strength of the synaptic weights, which can be viewed as a resource constraint for building and maintaining strong synapses. We note again that ridge regression is achieved as a special case when $a = 1$ and $b = \lambda$.

## 2.2 Quantifying differences between networks through the lens of decoding

We now turn our attention to the the problem of quantifying representational similarity between $\boldsymbol{X} \in \mathbb{R}^{M \times N_X}$ and $\boldsymbol{Y} \in \mathbb{R}^{M \times N_Y}$. As before, we specify a target vector $\boldsymbol{z} \in \mathbb{R}^M$ and optimize $\boldsymbol{w}$ as

specified in eq. (1). We optimize $v$ accordingly:

$$\underset{v}{\text{maximize}} \quad \frac{1}{M} z^\top Y v - \frac{1}{2} v^\top G(Y) v \tag{6}$$

yielding the solution $v^* = \frac{1}{M} G(Y)^{-1} Y^\top z$. We are interested in quantifying similarity between $X$ and $Y$ by comparing the behavior of these optimal decoders. A straightforward way to quantify the alignment between the decoded signals is to compute their inner product, which we call the *decoding similarity* (fig. 1):

$$\langle X w^*, Y v^* \rangle = w^{*\top} X^\top Y v^* = z^\top K_X K_Y z \tag{7}$$

where we have leveraged eq. (2) and the analogous closed form expression for $v^*$ while introducing the following normalized kernel similarity matrices:

$$K_X := \frac{1}{M} X G(X)^{-1} X^\top \quad \text{and} \quad K_Y := \frac{1}{M} Y G(Y)^{-1} Y^\top. \tag{8}$$

Note that we have suppressed the dependence of $K_X$ and $K_Y$ on the function $G(\cdot)$ to reduce notational clutter.

Equation (7) is a reasonable quantification of similarity between $X$ and $Y$ with respect to a fixed decoding task specified by $z \in \mathbb{R}^M$, and it may be a fruitful approach for hypothesis-driven comparisons of neural systems. We comment further on this possibility in our discussion. On the other hand, many neural representations support a variety of downstream behavioral tasks. For example, features extracted in early stages of visual processing (edges and textures) can be used to support many visual tasks such as object classification, segmentation, image compression, et cetera. How can eq. (7) be adapted to quantify the similarity between $X$ and $Y$ across more than one pre-specified decoding task? We explore three simple ideas.

**Option 1 — best case alignment.** An optimistic measure of representational similarity between networks is to find the decoding task $z \in \mathbb{R}^M$ that results in a maximal decoder alignment. Formally,

$$\max_{\|z\|_2=1} \langle X w^*, Y v^* \rangle = \max_{\|z\|_2=1} z^\top K_X K_Y z \tag{9}$$

where the constraint that $\|z\|_2 = 1$ is needed to keep the maximal value finite.

**Option 2 — worst case alignment.** An alternative is to find the task that minimizes alignment:

$$\min_{\|z\|_2=1} \langle X w^*, Y v^* \rangle = \min_{\|z\|_2=1} z^\top K_X K_Y z \tag{10}$$

which is obviously the pessimistic counterpart to option 1 above.

**Option 3 — average case alignment.** Instead of maximizing or minimizing over $z$, we can approach the problem by averaging over a distribution of decoding tasks. Formally, we can quantify alignment in terms of the *average decoding similarity*:

$$\mathbb{E} \langle X w^*, Y v^* \rangle \tag{11}$$

where the expectation is taken with respect to $z \sim P_z$.

In the next subsection, we show that the third option is closely related to several existing neural representational similarity measures. However, all three quantities are potentially of interest and they are easy to compute as we document below in propositions 1 and 2.

**Proposition 1.** *The best case and worst case decoding alignment scores, eqs.* (9) *and* (10)*, are respectively given by the largest and smallest eigenvalues of:*

$$\frac{1}{2} \left( K_X K_Y + K_Y K_X \right) \tag{12}$$

**Proposition 2.** *The average decoding similarity, eq.* (11)*, is given by:*

$$\mathbb{E} \langle X w^*, Y v^* \rangle = \text{Tr} \left( K_X K_z K_Y \right), \tag{13}$$

*where* $K_z := \mathbb{E}[z z^\top]$ *represents a kernel matrix for the targets.*

| Similarity measure | $a$ | $b$ |
|---|---|---|
| Linear CKA | 0 | $b$ |
| GULP | 1 | $\lambda$ |
| CCA | 1 | 0 |
| ENSD | 0 | $\frac{1}{M}\frac{\mathrm{Tr}[C_X^2]}{\mathrm{Tr}[C_X]}$ |

*Table 1:* Existing similarity measures are equivalent to the decoding score in eq. (7) or the decoding distance in eq. (16) for different choices of $a$ and $b$ in eq. (5).

*Proof of proposition 1.* Let $\boldsymbol{A} \in \mathbb{R}^{M \times M}$ be any matrix, potentially non-symmetric. We can express $\boldsymbol{A}$ as a sum of a symmetric and skew symmetric matrix, $\boldsymbol{A} = \boldsymbol{B} + \boldsymbol{C}$, where:

$$\boldsymbol{B} = \frac{1}{2}\left(\boldsymbol{A} + \boldsymbol{A}^\top\right) \qquad \text{and} \qquad \boldsymbol{C} = \frac{1}{2}\left(\boldsymbol{A} - \boldsymbol{A}^\top\right) \tag{14}$$

For any $\boldsymbol{z} \in \mathbb{R}^M$, we have $\boldsymbol{z}^\top \boldsymbol{A} \boldsymbol{z} = \boldsymbol{z}^\top \boldsymbol{B} \boldsymbol{z} + \boldsymbol{z}^\top \boldsymbol{C} \boldsymbol{z}$. It is easy to verify that $\boldsymbol{z}^\top \boldsymbol{C} \boldsymbol{z}$ equals zero. Thus, $\boldsymbol{z}^\top \boldsymbol{A} \boldsymbol{z} = \boldsymbol{z}^\top \boldsymbol{B} \boldsymbol{z}$, and maximizing over $\boldsymbol{z}$ subject to $\|\boldsymbol{z}\|_2$ yields the largest eigenvalue of $\boldsymbol{B}$ (since it is symmetric). Likewise, minimizing yields the smallest eigenvalue of $\boldsymbol{B}$. Substituting $\boldsymbol{A} = \boldsymbol{K}_X \boldsymbol{K}_Y$ into this argument proves the proposition. $\qquad\square$

*Proof of proposition 2.* Using eq. (7), we have:

$$\mathbb{E}\langle \boldsymbol{X}\boldsymbol{w}^*, \boldsymbol{Y}\boldsymbol{v}^*\rangle = \mathbb{E}[\boldsymbol{z}^\top \boldsymbol{K}_X \boldsymbol{K}_Y \boldsymbol{z}].$$

Using the cyclic property of the trace operator to manipulate this further, we get:

$$\boldsymbol{z}^\top \boldsymbol{K}_X \boldsymbol{K}_Y \boldsymbol{z} = \mathrm{Tr}(\boldsymbol{K}_X \boldsymbol{z}\boldsymbol{z}^\top \boldsymbol{K}_Y).$$

Taking the expectation with respect to $\boldsymbol{z}$ and using linearity of trace yields eq. (13). $\qquad\square$

## 3 Interpretations of CKA and related measures

Intuitively, proposition 2 says that the expected inner product between optimal linear readouts is equal to a weighted matrix inner product between kernel matrices $\boldsymbol{K}_X$ and $\boldsymbol{K}_Y$ (with respect to a positive semi-definite weighting matrix $\boldsymbol{K}_z$, which of course depends on the distribution of decoding targets $\boldsymbol{z}$). Such kernel matrices and inner products are important for several methods of quantifying representational similarity. We can therefore leverage this result to provide new interpretations of several distance measures. The most basic idea is to consider a distribution over $\boldsymbol{z}$ which satisfies $\boldsymbol{K}_z = \boldsymbol{I}$. This leads us to measure distances using standard Euclidean geometry, which we state as the following corollary to proposition 2. We explore a relaxation of this assumption and the limit as the number of input samples $M \to \infty$ in appendix C.

**Corollary 1.** *For any distribution over decoding targets satisfying $\boldsymbol{K}_z = \boldsymbol{I}$, the Frobenius inner product between normalized kernel matrices $\boldsymbol{K}_X$ and $\boldsymbol{K}_Y$ is equal to the average decoding similarity:*

$$\mathbb{E}\langle \boldsymbol{X}\boldsymbol{w}^*, \boldsymbol{Y}\boldsymbol{v}^*\rangle = \mathrm{Tr}[\boldsymbol{K}_X \boldsymbol{K}_Y] \tag{15}$$

*Further, when $\boldsymbol{K}_z = \boldsymbol{I}$, the (squared) average decoding distance is equal to the (squared) Euclidean distance between $\boldsymbol{K}_X$ and $\boldsymbol{K}_Y$:*

$$\mathbb{E}\|\boldsymbol{X}\boldsymbol{w}^* - \boldsymbol{Y}\boldsymbol{v}^*\|_2^2 = \|\boldsymbol{K}_X - \boldsymbol{K}_Y\|_F^2. \tag{16}$$

This corollary can be used to unify several neural representational similarity measures, each corresponding to a different choice of the penalty function $\boldsymbol{G}(\boldsymbol{X})$, Table 1. Further, it yields an interpretation of each measure in terms of the expected overlap or difference in decoding readouts between networks.

**Linear CKA**   First, for $b > 0$, the following choice of $\boldsymbol{G}(\boldsymbol{X})$ yields the linear CKA score [6, 22]:

$$\boldsymbol{G}(\boldsymbol{X}) = b\boldsymbol{I} \quad \Rightarrow \quad \frac{\mathbb{E}\langle \boldsymbol{X}\boldsymbol{w}^*, \boldsymbol{Y}\boldsymbol{v}^* \rangle}{\sqrt{\mathbb{E}\langle \boldsymbol{X}\boldsymbol{w}^*, \boldsymbol{X}\boldsymbol{w}^* \rangle \mathbb{E}\langle \boldsymbol{Y}\boldsymbol{v}^*, \boldsymbol{Y}\boldsymbol{v}^* \rangle}} = \frac{\mathrm{Tr}[\boldsymbol{K}_X \boldsymbol{K}_Y]}{\|\boldsymbol{K}_X\|_F \|\boldsymbol{K}_Y\|_F} = \mathrm{CKA}(\boldsymbol{X}, \boldsymbol{Y})$$

where we have normalized the expected inner product to obtain a similarity score ranging between 0 and 1. Note that the centering step of CKA is taken care of by our assumption, stated at the beginning of section 2, that the columns of $\boldsymbol{X}$ and $\boldsymbol{Y}$ are preprocessed to have mean zero. Thus, linear CKA be interpreted as a normalized average decoding similarity with quadratic weight regularization.

**GULP distance**   The CKA score is closely related to Euclidean distance, so it is not surprising that we can extend our analysis to other methods that utilize this distance. For example, the sample GULP distance, proposed by [3], is the equal to the average decoding distance with a regularization parameter $\lambda > 0$. In particular,

$$\boldsymbol{G}(\boldsymbol{X}) = \boldsymbol{C}_X + \lambda\boldsymbol{I} \quad \Rightarrow \quad \mathbb{E}\|\boldsymbol{X}\boldsymbol{w}^* - \boldsymbol{Y}\boldsymbol{v}^*\|_2^2 = \|\boldsymbol{K}_X - \boldsymbol{K}_Y\|_F^2 = \mathrm{GULP}_\lambda^2(\boldsymbol{X}, \boldsymbol{Y})$$

yields the plug-in estimate of the squared GULP distance [3, section 3]. We therefore see that the Euclidean distance between normalized kernel matrices can be interpreted as equal to the average decoding distance, or also as an approximation of the population formulation of the GULP distance established in [3].

**Canonical correlation analysis (CCA)**   CCA is another method that has been used to compare neural representations that fits nicely into our framework [29, 36]. When the two networks have the same dimension, i.e. $N = N_X = N_Y$, the output of CCA is a sequence of $N$ canonical correlation coefficients $1 \geq \rho_1 \geq \cdots \geq \rho_N \geq 0$. The average squared canonical coefficients is sometimes used as a measure of similarity,[1] and Kornblith et al. [22] showed this to be related to the linear CKA score on whitened representations. In particular, assuming the covariance matrices $\boldsymbol{C}_X$ and $\boldsymbol{C}_Y$ are invertible, we have:

$$\boldsymbol{G}(\boldsymbol{X}) = \boldsymbol{C}_X \quad \Rightarrow \quad \frac{\mathbb{E}\langle \boldsymbol{X}\boldsymbol{w}^*, \boldsymbol{Y}\boldsymbol{v}^* \rangle}{\sqrt{\mathbb{E}\langle \boldsymbol{X}\boldsymbol{w}^*, \boldsymbol{X}\boldsymbol{w}^* \rangle \mathbb{E}\langle \boldsymbol{Y}\boldsymbol{v}^*, \boldsymbol{Y}\boldsymbol{v}^* \rangle}} = \frac{1}{N}\sum_{i=1}^{N} \rho_i^2 \tag{17}$$

In practice, it is often useful to incorporate regularization into CCA because the covariance matrices $\boldsymbol{C}_X$ and $\boldsymbol{C}_Y$ may be ill-conditioned. As mentioned above, the regularization used in GULP distance corresponds to choosing $\boldsymbol{G}(\boldsymbol{X}) = \boldsymbol{C}_X + \lambda\boldsymbol{I}$. A similar approach proposed in [36] is to use $\boldsymbol{G}(\boldsymbol{X}) = (1-\alpha)\boldsymbol{C}_X + \alpha\boldsymbol{I}$ for a hyperparameter $0 \leq \alpha \leq 1$. This effectively allows for a continuous interpolation between a CCA-based similarity score and linear CKA.

**Effective Number of Shared Dimensions (ENSD)**   Finally, we note a connection to recent work by Giaffar et al. [15], who studied the following quantity, which they call the ENSD:

$$\mathrm{ENSD}(\boldsymbol{X}, \boldsymbol{Y}) = \frac{\mathrm{Tr}[\boldsymbol{X}\boldsymbol{X}^\top]\,\mathrm{Tr}[\boldsymbol{Y}\boldsymbol{Y}^\top]\,\mathrm{Tr}[\boldsymbol{X}\boldsymbol{X}^\top\boldsymbol{Y}\boldsymbol{Y}^\top]}{\mathrm{Tr}[\boldsymbol{X}\boldsymbol{X}^\top\boldsymbol{X}\boldsymbol{X}^\top]\,\mathrm{Tr}[\boldsymbol{Y}\boldsymbol{Y}^\top\boldsymbol{Y}\boldsymbol{Y}^\top]} \tag{18}$$

This is simply a rescaling of the inner product, $\mathrm{Tr}[\boldsymbol{X}\boldsymbol{X}^\top\boldsymbol{Y}\boldsymbol{Y}^\top]$ and therefore easily fits within our framework using the following choice for $\boldsymbol{G}(\boldsymbol{X})$:

$$\boldsymbol{G}(\boldsymbol{X}) = \frac{\mathrm{Tr}[\boldsymbol{C}_X^2]}{\mathrm{Tr}[\boldsymbol{C}_X]}\boldsymbol{I} \quad \Rightarrow \quad \mathbb{E}\langle \boldsymbol{X}\boldsymbol{w}^*, \boldsymbol{Y}\boldsymbol{v}^* \rangle = \mathrm{ENSD}(\boldsymbol{X}, \boldsymbol{Y}) \tag{19}$$

One of the most interesting features of ENSD is it's connection to the **participation ratio**, $\mathcal{R}$:

$$\mathcal{R}(\boldsymbol{C}_X) = \frac{(\mathrm{Tr}[\boldsymbol{C}_X])^2}{\mathrm{Tr}[\boldsymbol{C}_X^2]} = \mathrm{ENSD}(\boldsymbol{X}, \boldsymbol{X}) \tag{20}$$

which is used within physics and computational neuroscience (e.g. [14, 7, 31]) as a continuous analogue to the rank of a matrix, and is a measure of the effective dimensionality of the data matrix

---

[1]In the statistics literature, this summary statistic is sometimes called *Yanai's generalized generalized coefficient of determination* [30, 38].

$\boldsymbol{X}$. In particular, one can show that $1 \leq \mathcal{R}(\boldsymbol{C}_X) \leq \text{rank}(\boldsymbol{C}_X)$, and that these two inequalities respectively saturate when $\boldsymbol{C}_X$ has only one nonzero eigenvalue and when all nonzero eigenvalues are equal. It is also interesting to note that the *inverse* participation ratio can be captured by average decoding similarity under a very simple choice for $\boldsymbol{G}(\boldsymbol{X})$:

$$\boldsymbol{G}(\boldsymbol{X}) = \text{Tr}[\boldsymbol{C}_X]\boldsymbol{I} \quad \Rightarrow \quad \mathbb{E}\langle \boldsymbol{X}\boldsymbol{w}^*, \boldsymbol{X}\boldsymbol{w}^*\rangle = \mathbb{E}\|\boldsymbol{X}\boldsymbol{w}^*\|^2 = \frac{1}{\mathcal{R}(\boldsymbol{C}_X)} \tag{21}$$

We reiterate that all of the results enumerated above depend on choosing a distribution over decoding targets which satisfies $\boldsymbol{K}_z = \boldsymbol{I}$. This enforces the decoding targets $z_1, \ldots, z_M$ to be uncorrelated random variables with unit variance. The unit variance constraint is analogous to the constraint that $\|\boldsymbol{z}\|_2 = 1$ that we imposed when maximizing or minimizing over $\boldsymbol{z}$ in eqs. (9) and (10). Indeed, scaling the variance of the $\boldsymbol{z}_i$'s simply scales the magnitude of $\mathbb{E}\langle \boldsymbol{X}\boldsymbol{w}^*, \boldsymbol{Y}\boldsymbol{v}^*\rangle$. Thus, setting the variance of each sample to one (or some other constant) is reasonable. The constraint that $\mathbb{E}[z_i z_j] = 0$ for all $i \neq j$ may be relaxed, as we discuss in appendix C.

## 4   Relating the Shape of Neural Representations to Decoding

In corollary 1 we saw that the Euclidean distance and Frobenius inner product between $\boldsymbol{K}_X$ and $\boldsymbol{K}_Y$ enjoys a nice interpretation in terms of (average) decoder similarity. This does not provide a completely satisfactory answer of our original question: What does the *geometry* of a neural representation imply about its function (i.e. decoder performance)? As discussed in section 1, comparing neural representations through linear kernel matrices, $\boldsymbol{K}_X$ and $\boldsymbol{K}_Y$, is motivated from geometric principles—namely, their invariance to orthogonal transformations. Indeed, for any orthogonal matrix $\boldsymbol{R}$, one can verify that:

$$\boldsymbol{G}(\boldsymbol{X}\boldsymbol{R})^{-1} = \boldsymbol{R}^\top \boldsymbol{G}(\boldsymbol{X})^{-1}\boldsymbol{R} \tag{22}$$

whenever $\boldsymbol{G}(\boldsymbol{X})$ is parameterized as in eq. (4). Therefore, then the transformation $\boldsymbol{X} \mapsto \boldsymbol{X}\boldsymbol{R}$ leaves the whitened linear kernel matrix unchanged. That is, if $\boldsymbol{X}' = \boldsymbol{X}\boldsymbol{R}$, we have:

$$\boldsymbol{K}_X = \tfrac{1}{M}\boldsymbol{X}\boldsymbol{G}(\boldsymbol{X})^{-1}\boldsymbol{X}^\top = \tfrac{1}{M}\boldsymbol{X}\boldsymbol{R}\boldsymbol{G}(\boldsymbol{X}\boldsymbol{R})^{-1}\boldsymbol{R}^\top\boldsymbol{X}^\top = \boldsymbol{K}_{X'}. \tag{23}$$

The **Procrustes shape distance**, $\mathcal{P}(\cdot, \cdot)$, is another popular method which is invariant to orthogonal transformations [36, 9]. This approach connects to a broader and decades-old literature on *shape analysis* [10, 18] whose applications include anatomical comparisons across biological species [12], and analysis of planar curves such as handwriting data [33]. If the two neural responses are mean-centered and have the same dimensionality, then the Procrustes distance is simply the minimal Euclidean distance obtained by rotating and reflecting one set of responses onto the other. That is, when $\boldsymbol{X} \in \mathbb{R}^{M \times N_X}$ and $\boldsymbol{Y} \in \mathbb{R}^{M \times N_Y}$ and we have $N = N_X = N_Y$, the Procrustes distance is:

$$\mathcal{P}(\boldsymbol{X}, \boldsymbol{Y}) = \min_{\boldsymbol{R} \in \mathcal{O}(N)} \|\boldsymbol{X} - \boldsymbol{Y}\boldsymbol{R}\|_F \tag{24}$$

When $N_X \neq N_Y$, a straightforward generalization of the Procrustes distance is to zero pad column-wise so that the matrix dimensions match. Geometrically, this can be interpreted as embedding the lower-dimensional point cloud into the higher-dimensional space and optimizing $\boldsymbol{R}$ within this space to align the point clouds.

The fact that Procrustes distance is defined in terms of an optimal alignment transformation is appealing both intuitively and because it suggests avenues to generalize the method, such as using permutations (instead of orthogonal transformations) to align representations [19]. Additionally, some empirical results have highlighted cases where Procrustes performs "better" than linear CKA [9, 5]. Rigorously defining and benchmarking what it means to be "better" in this context remains an open discussion [21], but it is nonetheless of interest to understand what Procrustes distance and associated definitions of *shape* imply about neural decoding.

In appendix A.1 we prove the following result, which states that the Procrustes distance between appropriately normalized representations constrains the average decoding distance, or equivalently, that the average decoding distance bounds the Procrustes distance from above and below.

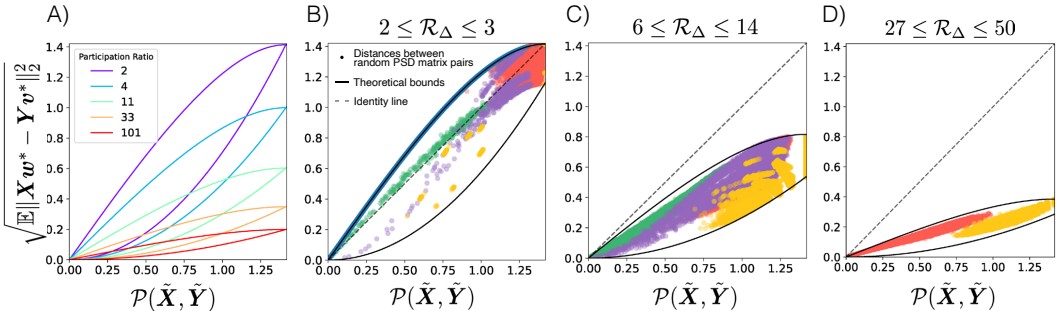

*Figure 2:* Bounds in eq. (25) plotted (solid lines) for varying participation ratio $\mathcal{R}_\Delta$ with $\alpha = \beta = 1$. A) Allowed regions (between the solid curves of like color) of Procrustes distance and expected Euclidean distance between decoded signals for different values of participation ratio $\mathcal{R}_\Delta$. B–D) Allowed regions for particular $\mathcal{R}_\Delta$ intervals (black solid lines) populated with calculated distances between pairs of randomly sampled positive semi-definite (PSD) matrices of size $50 \times 50$, subsampled to those that have $\mathcal{R}_\Delta$ in the particular interval (colored points). Different colors represent different random ensembles of positive semi-definite matrix pairs, which are described in appendix B.

**Proposition 3.** *Defining normalization constants $\alpha = \mathrm{Tr}\,\boldsymbol{K}_X$ and $\beta = \mathrm{Tr}\,\boldsymbol{K}_Y$, we have an upper and lower bound on the Procrustes distance:*

$$(\alpha + \beta) - \sqrt{(\alpha+\beta)^2 - \mathcal{R}_\Delta \mathbb{E}\|\boldsymbol{X}\boldsymbol{w}^* - \boldsymbol{Y}\boldsymbol{v}^*\|_2^2} \leq \mathcal{P}(\tilde{\boldsymbol{X}},\tilde{\boldsymbol{Y}})^2 \leq \sqrt{\mathcal{R}_\Delta \mathbb{E}\|\boldsymbol{X}\boldsymbol{w}^* - \boldsymbol{Y}\boldsymbol{v}^*\|_2^2} \quad (25)$$

*where $\mathcal{R}_\Delta = \mathcal{R}(\boldsymbol{K}_X - \boldsymbol{K}_Y)$ is the participation ratio of the matrix difference $\boldsymbol{K}_X - \boldsymbol{K}_Y$. Equivalently, we may rewrite eq. (25) to find*

$$\frac{1}{\mathcal{R}_\Delta}\mathcal{P}(\tilde{\boldsymbol{X}},\tilde{\boldsymbol{Y}})^4 \leq \mathbb{E}\|\boldsymbol{X}\boldsymbol{w}^* - \boldsymbol{Y}\boldsymbol{v}^*\|_2^2 \leq \frac{2(\alpha+\beta)\mathcal{P}(\tilde{\boldsymbol{X}},\tilde{\boldsymbol{Y}})^2 - \mathcal{P}(\tilde{\boldsymbol{X}},\tilde{\boldsymbol{Y}})^4}{\mathcal{R}_\Delta}, \quad (26)$$

*where we have defined normalized transformations $\tilde{\boldsymbol{X}}$ and $\tilde{\boldsymbol{Y}}$ in terms of the transformation $\boldsymbol{G}(\cdot)$:*

$$\tilde{\boldsymbol{X}} := \frac{1}{\sqrt{M}}\boldsymbol{X}\boldsymbol{G}(\boldsymbol{X})^{-1/2} \quad and \quad \tilde{\boldsymbol{Y}} := \frac{1}{\sqrt{M}}\boldsymbol{Y}\boldsymbol{G}(\boldsymbol{Y})^{-1/2}.$$

*Note that $\tilde{\boldsymbol{X}}\tilde{\boldsymbol{X}}^\top = \boldsymbol{K}_X$ and $\tilde{\boldsymbol{Y}}\tilde{\boldsymbol{Y}}^\top = \boldsymbol{K}_Y$.*

We derive these bounds by leveraging the equivalence between the Procrustes distance between normalized representations $\mathcal{P}(\tilde{\boldsymbol{X}},\tilde{\boldsymbol{Y}})$ and a distance metric on positive semi-definite kernel matrices, the *Bures distance* [16].

We recall that the choice of function $\boldsymbol{G}(\cdot)$ in the decoding problem eq. (1) (which determines optimal readout weights $\boldsymbol{w}^*$ and $\boldsymbol{v}^*$) will set the normalization transformation of the representations $\tilde{\boldsymbol{X}}$ and $\tilde{\boldsymbol{Y}}$. When $\boldsymbol{G}(\cdot)$ is chosen such that $\mathrm{Tr}\,\boldsymbol{K}_X = \mathrm{Tr}\,\boldsymbol{K}_Y = 1$,[2] we can plot these bounds with $\alpha = \beta = 1$ for various values of participation ratio $\mathcal{R}_\Delta$ (fig. 2).

Both upper and lower bounds are saturated (appendix A.1), and reveal an allowed region of both Procrustes and average decoding distance as a function of the participation ratio $\mathcal{R}_\Delta$. Under the $\alpha = \beta = 1$ normalization, the Procrustes distance may range between 0 and $\sqrt{2}$, and the average decoding distance may take values between 0 and $\frac{2}{\sqrt{\mathcal{R}_\Delta}}$, which is reflected in fig. 2.

The bounds in eq. (25) and eq. (26) reveal an imperfect connection between the Procrustes distance and the average decoding distance, such as the GULP distance. We recall that every choice of readout weight normalization $\boldsymbol{G}(\boldsymbol{X})$ leads to a measure of representational distance $\mathbb{E}\|\boldsymbol{X}\boldsymbol{w}^* - \boldsymbol{Y}\boldsymbol{v}^*\|_2^2$, and that there is an associated Procrustes distance on the normalized representations $\tilde{\boldsymbol{X}}$ and $\tilde{\boldsymbol{Y}}$,

---

[2]For example $\boldsymbol{G}(\boldsymbol{X}) = N_X \boldsymbol{C}_X$ or the CKA choice from above with $b = \mathrm{Tr}[\boldsymbol{C}_X]$. This is not a requirement for these bounds to hold, but simply a choice of normalization for the sake of plotting.

$\mathcal{P}(\tilde{\boldsymbol{X}}, \tilde{\boldsymbol{Y}})$. These bounds tell us that the average decoding distance $\mathbb{E}\|\boldsymbol{X}\boldsymbol{w}^* - \boldsymbol{Y}\boldsymbol{v}^*\|_2^2$ and this Procrustes distance constrain each other in a very particular way. Namely, when the Procrustes distance between normalized representations is small, then the average decoding distance must also be small.[3] However, the converse is only necessarily true when the participation ratio of the difference of the kernel matrices $\boldsymbol{K}_X - \boldsymbol{K}_Y$ is also small (e.g. fig. 2B). Indeed, as the dimensionality of $\boldsymbol{K}_X - \boldsymbol{K}_Y$ increases, we see that the Procrustes distance between normalized representations can be large, but the average decoding distance can be relatively small, with the effect becoming more exaggerated as $\mathcal{R}_\Delta$ becomes large (fig. 2D). In appendix A.1 we show that for $\alpha = \beta = 1$, the minimum possible participation ratio $\mathcal{R}_\Delta$ is 2. Therefore, fig. 2 also illustrates another interesting phenomenon, namely that the upper left-hand corner of these plots is impossible to populate. That is, in the case of large expected difference in decoded signals, one can never simultaneously measure a small Procrustes distance between the normalized representations. Thus we observe quantitatively, as may be intuitive, that the Procrustes distance offers a more strict notion of geometric dissimilarity than the average decoding distance.

## 5 Discussion

Understanding meaningful ways to measure similarities between neural representations is clearly a very complex problem. The literature demonstrates a proliferation of different techniques [20, 34], but an underdeveloped understanding of how these methods relate to each other. Less still is known about how representational similarity measures interact with functional similarity or the amount and type of information that is decodable from the representation. This paper presents a theoretical framework centered around the linear decoding of information from representations, which allows us to understand some existing popular methods for measuring representational similarity as average decoding similarity or average decoding distance with different choices of weight regularization. These connections relied on averaging the decoding similarity or decoding distance over a distribution of decoding targets $\boldsymbol{z}$ with $\mathbb{E}[\boldsymbol{z}\boldsymbol{z}^\top] = \boldsymbol{I}$. In the future it could be interesting to explore modifying these assumptions. For instance, instead of maximizing, minimizing, or taking the expectation over decoding targets, are there potentially interesting sets of fixed decoding targets that make sense when comparing networks in particular contexts? Furthermore, we focused on linear regression as a decoding method; a potentially interesting line of future work could be to extend this framework to linear classifiers (e.g. support vector machines or multi-class logistic regression). Lastly, in this paper we considered quantifying representational similarity across a finite set of $M$ stimulus conditions. However, these finite-dimensional approaches can be framed as approximations or estimators for a population version of the problem. For example, the framework in [28] for the Procrustes distance, or [3] for the GULP distance (the plugin estimator of which is a special case of our framework as we saw above). We outline a framing of this perspective in appendix C, but a rigorous exploration of the $M \to \infty$ regime and an analysis of the behavior of estimators for similarity scores in the limited sample regime could be an important topic of future work.

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

# A    Appendix

## A.1    Bounds on the Procrustes distance in terms of the Euclidean distance

Here we derive upper and lower bounds on the Procrustes distance between neural representations $X \in \mathbb{R}^{M \times N_X}$ and $Y \in \mathbb{R}^{M \times N_Y}$ in terms of the Euclidean distance between linear kernel matrices $K_X = XX^\top$ and $K_Y = YY^\top$. This result is applied in the main text to relate the expected Euclidean distance between decoded signals $\mathbb{E}\|Xw^* - Yv^*\|_2^2 = \|\tilde{X}\tilde{X}^\top - \tilde{Y}\tilde{Y}^\top\|_F^2$ to the Procrustes distance $\mathcal{P}(\tilde{X}, \tilde{Y})$. Tildes are omitted in this section for clarity, but we note that this result is applied in the main text to the normalized representations $\tilde{X}$ and $\tilde{Y}$.

We will make use of the equivalence between the Procrustes distance between representation matrices $X$ and $Y$ and a notion of distance on the space of positive semi-definite kernel matrices $K_X = XX^\top$ and $K_Y = YY^\top$ called the Bures distance, defined as

$$d_B(K_X, K_Y) = \sqrt{\operatorname{Tr}[K_X] + \operatorname{Tr}[K_Y] - 2\operatorname{Tr}\left[\left(K_X^{1/2} K_Y K_X^{1/2}\right)^{1/2}\right]}. \tag{27}$$

The third trace term on the right hand side of eq. (27) is often called the *fidelity*, defined as

$$\mathcal{F}(K_X, K_Y) = \operatorname{Tr}\left[\left(K_X^{1/2} K_Y K_X^{1/2}\right)^{1/2}\right]. \tag{28}$$

### A.1.1    Lower bound

For the lower bound, we are inspired by the Fuchs-van-de-Graaf inequalites that are used in quantum information theory to assert an approximate equivalence between two measures of quantum state similarity, the *trace distance* and the *fidelity*. This approach is relevant from a mathematical perspective for our purposes, since quantum states are represented by positive semi-definite matrices normalized to have trace 1. Here, we use a similar approach as that often used to derive the Fuchs-van de Graaf inequalites to instead relate the Euclidean distance and the Bures distance on positive semidefinite matrices, while also relaxing the trace 1 normalization.

We will make use of the following identity.

**Lemma 1.**
$$\|\alpha u u^\dagger - \beta v v^\dagger\|_* = \sqrt{(\alpha + \beta)^2 - 4\alpha\beta|\langle u, v\rangle|^2} \tag{29}$$

*for all unit vectors $u, v$ and all non-negative real numbers $\alpha$ and $\beta$.*

*Proof.* First we recognize that the nuclear norm of a matrix can be calculated by summing the singular values of that matrix. Furthermore, if that matrix is Hermitian, the singular values are the absolute values of the eigenvalues. Define $M = \alpha u u^\dagger - \beta v v^\dagger$. Our matrix $M$ is Hermitian and has at most two eigenvalues, so we will look for expressions for these in terms of $\alpha$ and $\beta$. Note that if $u$ and $v$ are the same unit vector, then $M$ has rank 1, so we expect the eigenvalues will depend also on the inner product $\langle u, v\rangle$.

The eigenvectors of $M$ will be of the form $\psi = cu + dv$ for some scalar $c$ and $d$. By direct calculation:

$$\begin{aligned} M\psi &= \lambda\psi \\ M(cu + dv) &= (\alpha c + \alpha d\langle u, v\rangle)u - (\beta c\langle v, u\rangle + \beta d)v \\ &= \lambda(cu + dv) \end{aligned} \tag{30}$$

Therefore, we see that the eigenvalues must satisfy both

$$\lambda = \frac{\alpha c + \alpha d\langle u, v\rangle}{c} \quad \text{and} \quad \lambda = -\frac{\beta c\langle v, u\rangle + \beta d}{d} \tag{31}$$

Setting these equal to each other and solving the resulting quadratic for $c$ gives:

$$c = \frac{-d(\alpha + \beta) \pm d\sqrt{(\alpha + \beta)^2 - 4\alpha\beta|\langle v, u \rangle|^2}}{2\beta\langle v, u \rangle}. \tag{32}$$

So the two eigenvectors are proportional to

$$\psi_\pm = \left(\frac{(\alpha + \beta)}{2} \mp \frac{1}{2}\sqrt{(\alpha + \beta)^2 - 4\alpha\beta|\langle v, u \rangle|^2}\right)u - \beta\langle v, u \rangle v. \tag{33}$$

To find the eigenvalues, we combine eq. (31) and eq. (32), arriving at

$$\lambda_\pm = \frac{(\beta - \alpha)}{2} \pm \frac{1}{2}\sqrt{(\alpha + \beta)^2 - 4\alpha\beta|\langle v, u \rangle|^2}. \tag{34}$$

Finally, to calculate $\|\boldsymbol{M}\|_*$ we simply sum the magnitudes of these eigenvalues. By inspecting the term under the square root, we see that $\lambda_-$ is always negative. Therefore, $|\lambda_+| + |\lambda_-| = \lambda_+ - \lambda_-$, and we find

$$\|\boldsymbol{M}\|_1 = |\lambda_+| + |\lambda_-|$$
$$\implies \|\alpha uu^\dagger - \beta vv^\dagger\|_1 = \sqrt{(\alpha + \beta)^2 - 4\alpha\beta|\langle v, u \rangle|^2} \tag{35}$$

$\square$

We now find a lower bound on the Bures distance between kernel matrices $\boldsymbol{K}_X$ and $\boldsymbol{K}_Y$ in terms of the Euclidean distance between the same matrices, $\|\boldsymbol{K}_X - \boldsymbol{K}_Y\|_F^2$. There are many choices of $\boldsymbol{X}$ and $\boldsymbol{Y}$ multiply to the same positive semi-definite kernel matrices via $\boldsymbol{K}_X = \boldsymbol{X}\boldsymbol{X}^\top$ and $\boldsymbol{K}_Y = \boldsymbol{Y}\boldsymbol{Y}^\top$. Define real positive scalars $\alpha = \operatorname{Tr} \boldsymbol{X}\boldsymbol{X}^\top$ and $\beta = \operatorname{Tr} \boldsymbol{Y}\boldsymbol{Y}^\top$, and the unit vectors

$$\hat{x} = \frac{\operatorname{vec}(\boldsymbol{X})}{\|\operatorname{vec}(\boldsymbol{X})\|_2} \quad \text{and} \quad \hat{y} = \frac{\operatorname{vec}(\boldsymbol{Y})}{\|\operatorname{vec}(\boldsymbol{Y})\|_2} \tag{36}$$

where $\operatorname{vec}(\boldsymbol{X})$ is the vectorization linear transformation converting the $M \times N_X$-dimensional matrix $\boldsymbol{X}$ into an $MN_X$-dimensional vector. We will assume that the representations have been zero-padded such that $N_X = N_Y$.

By Uhlmann's theorem [35], there exists some choice of $\boldsymbol{X}$ and $\boldsymbol{Y}$, such that

$$|\langle \sqrt{\alpha}\hat{x}, \sqrt{\beta}\hat{y} \rangle| = \mathcal{F}(\boldsymbol{K}_X, \boldsymbol{K}_Y) \tag{37}$$

One can see this by writing the fidelity as a nuclear norm of $\boldsymbol{X}^\top\boldsymbol{Y}$ and considering the variational form of the nuclear norm as a maximization of the Hilbert-Schmidt inner product over unitary transformations (see [16] for a more explicit discussion of this).

Now use our identity from earlier with $u = \hat{x}$ and $v = \hat{y}$.

$$\|\alpha\hat{x}\hat{x}^\top - \beta\hat{y}\hat{y}^\top\|_1 = \sqrt{(\alpha + \beta)^2 - 4\alpha\beta|\langle \hat{x}, \hat{y} \rangle|^2}$$
$$= \sqrt{(\alpha + \beta)^2 - 4\mathcal{F}(\boldsymbol{K}_X, \boldsymbol{K}_Y)^2} \tag{38}$$

The operation that takes $\alpha\hat{x}\hat{x}^\top$ and $\beta\hat{y}\hat{y}^\top$ to $\boldsymbol{K}_X$ and $\boldsymbol{K}_Y$, respectively, is called the partial trace. By the monotonicity of the nuclear norm under partial tracing [35], we have:

$$\|\boldsymbol{K}_X - \boldsymbol{K}_Y\|_* \leq \|\alpha\hat{x}\hat{x}^\top - \beta\hat{y}\hat{y}^\top\|_* = \sqrt{(\alpha + \beta)^2 - 4\mathcal{F}(\boldsymbol{K}_X, \boldsymbol{K}_Y)^2}. \tag{39}$$

With equality when $\boldsymbol{K}_X$ and $\boldsymbol{K}_Y$ are rank-1, that is $\boldsymbol{K}_X = \alpha\hat{x}\hat{x}^\top$ and $\boldsymbol{K}_Y = \beta\hat{y}\hat{y}^\top$.

Lastly, we rewrite the nuclear norm $\|\boldsymbol{K}_X - \boldsymbol{K}_Y\|_*$ in terms of the Euclidean distance and the *participation ratio* of the matrix $\boldsymbol{K}_X - \boldsymbol{K}_Y$. The participation ratio of a matrix $\boldsymbol{A} \in \mathbb{R}^{M \times N}$ is defined as

$$\mathcal{R}(\boldsymbol{A}) = \frac{\|\boldsymbol{A}\|_1^2}{\|\boldsymbol{A}\|_F^2} = \frac{(\sum_i \sigma_i)^2}{\sum_i \sigma_i^2}, \tag{40}$$

so we have

$$\|\boldsymbol{K}_X - \boldsymbol{K}_Y\|_* = \sqrt{\mathcal{R}_\Delta} \|\boldsymbol{K}_X - \boldsymbol{K}_Y\|_F \tag{41}$$

where $\mathcal{R}_\Delta = \mathcal{R}(\tilde{\boldsymbol{X}}\tilde{\boldsymbol{X}}^\top - \tilde{\boldsymbol{Y}}\tilde{\boldsymbol{Y}}^\top)$. We can also use the definition of the Bures distance eq. (27) to replace $\mathcal{F}(\boldsymbol{K}_X, \boldsymbol{K}_Y) = \frac{1}{2}(\alpha + \beta - d_B(\boldsymbol{K}_X, \boldsymbol{K}_Y)^2)$. Making these substitutions into eq. (39), and solving for $d_B(\boldsymbol{K}_X, \boldsymbol{K}_Y)^2$, we find:

$$d_B(\boldsymbol{K}_X, \boldsymbol{K}_Y)^2 \geq (\alpha + \beta) - \sqrt{(\alpha + \beta)^2 - \mathcal{R}_\Delta \|\boldsymbol{K}_X - \boldsymbol{K}_Y\|_F^2}. \tag{42}$$

This bound is saturated when the inequalty in eq. (39) is an equality, or when $\boldsymbol{K}_X$ and $\boldsymbol{K}_Y$ are both rank 1.

For a bound that is independent of the participation ratio, we could replace $\mathcal{R}_\Delta$ with its ostensible minimal value of 1. When $\boldsymbol{K}_X$ and $\boldsymbol{K}_Y$ are normalized to have trace 1, we would then have

$$d_B(\boldsymbol{K}_X, \boldsymbol{K}_Y)^2 \geq 2 - \sqrt{4 - \|\boldsymbol{K}_X - \boldsymbol{K}_Y\|_F^2} \approx \frac{1}{4}\|\boldsymbol{K}_X - \boldsymbol{K}_Y\|_F^2. \tag{43}$$

However, in this case that $\text{Tr}\,\boldsymbol{K}_X = \text{Tr}\,\boldsymbol{K}_Y = 1$, we can actually arrive at a tighter lower bound than eq. (43), because we can say something interesting about the minimal participation ratio of matrices of the form $\boldsymbol{K}_X - \boldsymbol{K}_Y$. It turns out that under this normalization, the participation ratio is lower bounded by 2, with the minimum of 2 achieved when $\boldsymbol{K}_X$ and $\boldsymbol{K}_Y$ are rank 1 (also in this case the bound in eq. (42) will be saturated).

**Lemma 2.** *For positive semidefinite matrices $\boldsymbol{K}_X$ and $\boldsymbol{K}_Y$ with $\text{Tr}\,\boldsymbol{K}_X = \text{Tr}\,\boldsymbol{K}_Y$,*

$$\mathcal{R}(\boldsymbol{K}_X - \boldsymbol{K}_Y) \geq 2. \tag{44}$$

*Proof.* By noting that $\boldsymbol{K}_X - \boldsymbol{K}_Y$ is Hermitian, we rewrite eq. (40) as

$$\mathcal{R}(\boldsymbol{K}_X - \boldsymbol{K}_Y) = \frac{(\sum_i |\lambda_i|)^2}{\sum_i |\lambda_i|^2} \tag{45}$$

where $\lambda_i$ is the $i$th eigenvalue of the matrix difference $\boldsymbol{K}_X - \boldsymbol{K}_Y$. Next, we recognize that if $\text{Tr}\,\boldsymbol{K}_X = \text{Tr}\,\boldsymbol{K}_Y$, then $\text{Tr}(\boldsymbol{K}_X - \boldsymbol{K}_Y) = 0$ and therefore the eigenvalues of $\boldsymbol{K}_X - \boldsymbol{K}_Y$ sum to 0. The set of eigenvalues $\{\lambda_1, \lambda_2, ..., \lambda_M\}$ can then always be partitioned into a set of positive eigenvalues $\{\lambda_1^+, \lambda_2^+, ..., \lambda_p^+\}$, and a set of negative eigenvalues $\{\lambda_1^-, \lambda_2^-, ..., \lambda_q^-\}$, with $p + q = M$. The negative eigenvalues must balance the positive eigenvalues in magnitude, so we must have

$$\sum_{i=1}^p \lambda_i^+ = \sum_{j=1}^q |\lambda_j^-|. \tag{46}$$

We can rewrite eq. (45) in terms of sums over the positive and negative eigenvalue sets:

$$\mathcal{R}(\boldsymbol{K}_X - \boldsymbol{K}_Y) = \frac{4(\sum_{i=1}^p \lambda_i^+)^2}{\sum_{i=1}^p (\lambda_i^+)^2 + \sum_{j=1}^q |\lambda_j^-|^2}$$

$$= \frac{4\big[\sum_{i=1}^p (\lambda_i^+)^2 + \sum_{j\neq k}^p \lambda_j^+ \lambda_k^+\big]}{(\sum_{i=1}^p \lambda_i^+)^2 - \sum_{j\neq k}^p \lambda_j^+ \lambda_k^+ + (\sum_{i=1}^q |\lambda_i^-|)^2 - \sum_{j\neq k}^q |\lambda_j^-||\lambda_k^-|}$$

$$= \frac{4\big[\sum_{i=1}^p (\lambda_i^+)^2 + \sum_{j\neq k}^p \lambda_j^+ \lambda_k^+\big]}{2[(\sum_{i=1}^p (\lambda_i^+)^2) + \frac{1}{2}(\sum_{j\neq k}^p \lambda_j^+ \lambda_k^+ - \sum_{j\neq k}^q |\lambda_j^-||\lambda_k^-|)]}$$

$$= 2\left[\frac{\sum_{i=1}^p (\lambda_i^+)^2 + \sum_{j\neq k}^p \lambda_j^+ \lambda_k^+}{\sum_{i=1}^p (\lambda_i^+)^2 + \frac{1}{2}(\sum_{j\neq k}^p \lambda_j^+ \lambda_k^+ - \sum_{j\neq k}^q |\lambda_j^-||\lambda_k^-|)}\right] \tag{47}$$

By inspection, the term in brackets is $\geq 1$ (this can be seen by considering the signs and relative magnitudes of each of the summed terms). So we have

$$\mathcal{R}(\boldsymbol{K}_X - \boldsymbol{K}_Y) \geq 2. \tag{48}$$

$\square$

The inequality eq. (44) is saturated when $\boldsymbol{K}_X - \boldsymbol{K}_Y$ is rank 2 and thus only has two equal magnitude and opposite signed eigenvalues, as can be seen from the first line of eq. (47). We can also use eq. (34) to see that when $\boldsymbol{K}_X$ and $\boldsymbol{K}_Y$ are rank 1 and trace 1, we have

$$\mathcal{R}(\boldsymbol{K}_X - \boldsymbol{K}_Y) = \frac{\|\boldsymbol{K}_X - \boldsymbol{K}_Y\|_1^2}{\|\boldsymbol{K}_X - \boldsymbol{K}_Y\|_F^2} = \frac{(\alpha + \beta)^2 - 4\alpha\beta|\langle v, u\rangle|^2}{\frac{1}{2}[(\alpha+\beta)^2 - 4\alpha\beta|\langle v,u\rangle|^2]} = 2. \tag{49}$$

For $\boldsymbol{K}_X$ and $\boldsymbol{K}_Y$ normalized to have trace 1, we can then state a tighter bound than eq. (43) using the minimal participation ratio in eq. (44). We have

$$d_B(\boldsymbol{K}_X, \boldsymbol{K}_Y)^2 \geq 2 - \sqrt{4 - 2\|\boldsymbol{K}_X - \boldsymbol{K}_Y\|_F^2}. \tag{50}$$

### A.1.2  Upper bound

We can bound the Bures distance using its variational form [2],

$$d_B^2(\boldsymbol{K}_X, \boldsymbol{K}_Y) = \min_{\boldsymbol{Q} \in \mathcal{O}(M)} \|\boldsymbol{K}_X^{1/2} - \boldsymbol{K}_Y^{1/2}\boldsymbol{Q}\|_F^2 \leq \|\boldsymbol{K}_X^{1/2} - \boldsymbol{K}_Y^{1/2}\|_F^2 \tag{51}$$

The Powers-Størmer inequality implies that $\|\boldsymbol{K}_X^{1/2} - \boldsymbol{K}_Y^{1/2}\|_F^2 \leq \|\boldsymbol{K}_X - \boldsymbol{K}_Y\|_*$, so

$$d_B^2(\boldsymbol{K}_X, \boldsymbol{K}_Y) \leq \|\boldsymbol{K}_X - \boldsymbol{K}_Y\|_*. \tag{52}$$

Expressing the nuclear norm in terms of the participation ratio of $\boldsymbol{K}_X - \boldsymbol{K}_Y$ using eq. (41), we have an upper bound on the Bures distance:

$$d_B^2(\boldsymbol{K}_X, \boldsymbol{K}_Y) \leq \sqrt{\mathcal{R}_\Delta}\|\boldsymbol{K}_X - \boldsymbol{K}_Y\|_F. \tag{53}$$

The inequality in eq. (51) is saturated when the optimal orthogonal transformation that aligns $\boldsymbol{K}_X^{1/2}$ and $\boldsymbol{K}_Y^{1/2}$ is the identity. This occurs when $\boldsymbol{K}_X$ and $\boldsymbol{K}_Y$ are simultaneously diagonalizable. The Powers-Størmer inequality in eq. (52) is saturated when $\boldsymbol{K}_X$ and $\boldsymbol{K}_Y$ have eigenvalues that only take values in $\{0, 1\}$.

## B  Figure 2 details

Figure 2 panels (B-D) show the allowed regions of Procrustes distance and expected Euclidean distance between decoded signals for representations normalized to have $\operatorname{Tr} \tilde{\boldsymbol{X}}\tilde{\boldsymbol{X}}^\top = \operatorname{Tr} \tilde{\boldsymbol{Y}}\tilde{\boldsymbol{Y}}^\top = 1$, with varying participation ratio $\mathcal{R}_\Delta$. We have populated the plots with points representing the respective distances between randomly sampled pairs of positive semi-definite matrices. These positive semi-definite matrices represent normalized kernel matrices $\tilde{\boldsymbol{X}}\tilde{\boldsymbol{X}}^\top$ and $\tilde{\boldsymbol{Y}}\tilde{\boldsymbol{Y}}^\top$. We have seen in the main text that the squared Euclidean distance between these kernel matrices is equivalent to the expected squared Euclidean distance between decoded signals, $\mathbb{E}\|\boldsymbol{X}\boldsymbol{w}^* - \boldsymbol{Y}\boldsymbol{v}^*\|_2^2 = \|\tilde{\boldsymbol{X}}\tilde{\boldsymbol{X}}^\top - \tilde{\boldsymbol{Y}}\tilde{\boldsymbol{Y}}^\top\|_F^2$, when the distribution of decoding targets satisfies $\mathbb{E}[\boldsymbol{z}\boldsymbol{z}^\top] = \boldsymbol{I}$. On the other hand, [16] demonstrated how the Procrustes distance $\mathcal{P}(\tilde{\boldsymbol{X}}, \tilde{\boldsymbol{Y}})$ is equivalent to the Bures distance eq. (27), $d_B(\tilde{\boldsymbol{X}}\tilde{\boldsymbol{X}}^\top, \tilde{\boldsymbol{Y}}\tilde{\boldsymbol{Y}}^\top)$. Therefore, both distances on these plots can be calculated from samples of positive semi-definite matrix pairs.

Each color of points in fig. 2 (B-D) represents a different random ensemble of $M \times M$ PSD matrices with $M = 50$, which are then binned into one of the three plots if the participation ratio $\mathcal{R}_\Delta$ lies in the respective range indicated above each plot panel.

- Pink points are generated by sampling eigenvalues for the two PSD matrices from a Dirichlet distribution with concentration parameters logarithmically spaced between $10^{-3}$ and $10^3$. Matrices of eigenvectors are then randomly sampled from the set of orthogonal matrices.

- Green points were generated by sampling a random matrix $\boldsymbol{A}$ of size $M \times r$ with each element drawn from the uniform distribution $\mathcal{U}[0, 1]$. A uniform random integer $r$ was selected between 1 and 24. A PSD matrix was generated by multiplying $\boldsymbol{A}\boldsymbol{A}^\top$. Another PSD matrix was then generated by adding a randomly weighted matrix of the same dimension with standard normal distributed entries $\boldsymbol{N}$ to $\boldsymbol{A}$, so $\boldsymbol{B} = \boldsymbol{A} + \epsilon\boldsymbol{N}$, where $\epsilon \sim \mathcal{U}[0, 1]$. The distance between $\frac{1}{\operatorname{Tr}\boldsymbol{A}\boldsymbol{A}^\top}\boldsymbol{A}\boldsymbol{A}^\top$ and $\frac{1}{\operatorname{Tr}\boldsymbol{B}\boldsymbol{B}^\top}\boldsymbol{B}\boldsymbol{B}^\top$ is then computed using these two distance metrics.

- Blue points are seen to only occupy fig. 2 (B), as these correspond to distances between matrices of rank 1 and trace 1. As we saw earlier in this appendix, in this case the participation ratio of the difference between these two PSD matrices is always 2. The rank 1 matrices were sampled using the same method as the green points above, but setting $r = 1$.

- Purple points were generated by sampling one matrix from a Wishart distribution $W_M(r, \boldsymbol{I})$, with degrees of freedom $r$ chosen as a uniformly random integer between 1 and 50. This matrix was normalized to have trace 1. The second matrix was generated by adding to the first matrix another matrix drawn from the Wishart distribution $W_M(n, \boldsymbol{I})$, with $n$ a randomly selected integer between 1 and 10. Both matrices are normalized to have trace 1.

- Yellow points represent distances between PSD matrices that are simultaneously diagonalizable. These were generated by sampling $M$ values uniformly between 0 and 1 for each matrix, but and setting these values to 0 if they fall below the threshold 0.6. The resultant values were then normalized to sum to 1, which became the eigenvalues for the two PSD matrices. A matrix of eigenvectors was then randomly selected from the set of orthogonal matrices.

## C  Generalizations and Interpretations as $M \to \infty$

In the main text, we considered quantifying representational similarity across a finite set of $M$ stimulus conditions. Further, when considering the average distance in decoding performance, we have assumed that $\mathbb{E}[\boldsymbol{z}\boldsymbol{z}^\top] = \boldsymbol{I}$. Here, we briefly discuss how to relax both constraints, leaving a full investigation to future work. Readers will need familiarity with the basic principles of kernel methods and gaussian processes in machine learning (see e.g. [32, 37]) to follow along with certain results in this section.

To begin, we must revisit and refine our theoretical framework developed in section 2. Under proposition 2, we treated the neural response matrices, $\boldsymbol{X}$ and $\boldsymbol{Y}$, as constants while we treated the decoder targets, $\boldsymbol{z}$, as a random variable. In this section we will treat $\boldsymbol{X}, \boldsymbol{Y}$ and $\boldsymbol{z}$ as joint random

variables, as has been done in prior work [28, 3]. Specifically, let $\mathcal{U}$ denote the space of possible stimulus inputs (e.g. the space of natural images or grammatically correct English sentences). We are interested in quantifying similarity between two neural networks, $f : \mathcal{U} \mapsto \mathbb{R}^{N_X}$ and $g : \mathcal{U} \mapsto \mathbb{R}^{N_Y}$, across this stimulus space. Let $P_{\boldsymbol{u}}$ denote a distribution with support on $\mathcal{U}$, and let $\boldsymbol{u}_1, \ldots, \boldsymbol{u}_M$ denote independent samples from $P_{\boldsymbol{u}}$. As before, we collect the network responses into matrices $\boldsymbol{X} \in \mathbb{R}^{M \times N_X}$ and $\boldsymbol{Y} \in \mathbb{R}^{M \times N_Y}$, which we assume to be mean centered, $\mathbb{E}[f(\boldsymbol{u})] = \mathbb{E}[g(\boldsymbol{u})] = \boldsymbol{0}$.

Finally, we define a decoding task as a function $\eta : \mathcal{U} \mapsto \mathbb{R}$. For $M$ finite samples (as considered in the main text), the decoder target is the vector with elements consisting of $\eta$ evaluated at each sampled input: $\boldsymbol{z} = [\eta(\boldsymbol{u}_1) \quad \ldots \quad \eta(\boldsymbol{u}_M)]^\top$. As in sections 3 and 4, we will be interested in characterizing the *average* decoder alignment over multiple tasks. In the main text we considered $\mathbb{E}_{\boldsymbol{z}} \langle \boldsymbol{X} \boldsymbol{w}^*, \boldsymbol{Y} \boldsymbol{v}^* \rangle$, but we would like to now generalize this to an expectation of an inner product between two *functions* over the input space. We consider $\eta$ to be drawn from a Gaussian process, $\eta \sim \mathcal{GP}(q)$, with a covariance operator $q : \mathcal{U} \times \mathcal{U} \mapsto \mathbb{R}$. Intuitively, $q$ defines the difficulty of the distribution over regression tasks by specifying smoothness with respect to the input space $\mathcal{U}$. For example, the popular squared exponential or radial basis function (RBF) kernel:

$$q(\boldsymbol{u}, \boldsymbol{u}') = \exp\left(-\frac{\|\boldsymbol{u} - \boldsymbol{u}'\|_2^2}{2\gamma}\right) \tag{54}$$

comes equipped with a length scale parameter $\gamma > 0$ that determines the smoothness of functions sampled $\eta \sim \mathcal{GP}(q)$. In the limit that $\gamma \to 0$ we get the Kronecker delta function:

$$\delta(\boldsymbol{u}, \boldsymbol{u}') = \begin{cases} 1 & \boldsymbol{u} = \boldsymbol{u}' \\ 0 & \boldsymbol{u} \neq \boldsymbol{u}' \end{cases} \tag{55}$$

In the remainder of this section, we show that the average decoding similarity converges to an expected inner product between kernels defined by the two neural systems. Specifically, let us define $k : \mathcal{U} \times \mathcal{U} \mapsto \mathbb{R}$ and $h : \mathcal{U} \times \mathcal{U} \mapsto \mathbb{R}$ as follows:

$$k(\boldsymbol{u}, \boldsymbol{u}') = f(\boldsymbol{u})^\top \boldsymbol{G}(f)^{-1} f(\boldsymbol{u}') \qquad \text{and} \qquad h(\boldsymbol{u}, \boldsymbol{u}') = g(\boldsymbol{u})^\top \boldsymbol{G}(g)^{-1} g(\boldsymbol{u}') \tag{56}$$

where we have generalized eq. (4) as:

$$\boldsymbol{G}(f) = a\mathbb{E}[f(\boldsymbol{u})f(\boldsymbol{u})^\top] + b\boldsymbol{I} \tag{57}$$

We denote by $L^2(\mathcal{U})$ the space of $L^2$-integrable functions from the set $\mathcal{U}$ to $\mathbb{R}$. Consider the integral operators $\mathcal{K} : L^2(\mathcal{U}) \to L^2(\mathcal{U})$ and $\mathcal{H} : L^2(\mathcal{U}) \to L^2(\mathcal{U})$ associated with the kernels in eq. (56). We define the functions $\eta_{\mathcal{K}}(\boldsymbol{u})$ and $\eta_{\mathcal{H}}(\boldsymbol{u})$ as these operators acting on the decoding task function $\eta(\boldsymbol{u})$:

$$[\mathcal{K}\eta](\boldsymbol{u}) = \int k(\boldsymbol{u}, \boldsymbol{u}')\eta(\boldsymbol{u}')p(\boldsymbol{u}')d\boldsymbol{u}' \equiv \eta_{\mathcal{K}}(\boldsymbol{u}) \tag{58}$$

$$[\mathcal{H}\eta](\boldsymbol{u}) = \int h(\boldsymbol{u}, \boldsymbol{u}')\eta(\boldsymbol{u}')p(\boldsymbol{u}')d\boldsymbol{u}' \equiv \eta_{\mathcal{H}}(\boldsymbol{u}) \tag{59}$$

The functions $\eta_{\mathcal{K}}(\boldsymbol{u})$ and $\eta_{\mathcal{H}}(\boldsymbol{u})$ are analogous to the vectors $\boldsymbol{X} \boldsymbol{w}^* = \boldsymbol{K}_X \in \mathbb{R}^M$ and $\boldsymbol{Y} \boldsymbol{v}^* = \boldsymbol{K}_Y \in \mathbb{R}^M$, and can be thought of as the 'decoded' functions. We will define a similarity score as the inner product:

$$S = \langle \eta_{\mathcal{K}}, \eta_{\mathcal{H}} \rangle_{L^2} = \int_{\mathcal{U}} \eta_{\mathcal{K}}(\boldsymbol{u})\eta_{\mathcal{H}}(\boldsymbol{u})p(\boldsymbol{u})d\boldsymbol{u} \tag{60}$$

Then we have, using the definitions in eq. (58) and eq. (59),

$$S = \int_{\mathcal{U}} \int_{\mathcal{U}} \int_{\mathcal{U}} k(\boldsymbol{u}, \boldsymbol{u}')\eta(\boldsymbol{u}')\eta(\boldsymbol{u}'')h(\boldsymbol{u}, \boldsymbol{u}'') \, p(\boldsymbol{u})d\boldsymbol{u} \, p(\boldsymbol{u}')d\boldsymbol{u}' \, p(\boldsymbol{u}'')d\boldsymbol{u}''. \tag{61}$$

This integral measures the similarity between two neural systems $f$ and $g$ for a particular choice of decoding task function $\eta$. However, following the main text, we would like to take the expectation over some ensemble of decoding task functions $\eta$.

$$\mathbb{E}_\eta[S] = \mathbb{E}_\eta \langle \eta_{\mathcal{K}}, \eta_{\mathcal{H}} \rangle_{L^2} \tag{62}$$

Assuming the conditions to invoke Fubini's theorem regarding changing the ordering of integration are met, we have

$$\mathbb{E}_\eta[S] = \int_{\mathcal{U}} \int_{\mathcal{U}} \int_{\mathcal{U}} k(\boldsymbol{u}, \boldsymbol{u}') q(\boldsymbol{u}', \boldsymbol{u}'') h(\boldsymbol{u}, \boldsymbol{u}'')\, p(\boldsymbol{u}) d\boldsymbol{u}\, p(\boldsymbol{u}') d\boldsymbol{u}'\, p(\boldsymbol{u}'') d\boldsymbol{u}''. \tag{63}$$

We now recognize this integral as this as the trace of a composition of integral operators,

$$\mathbb{E}_\eta[S] = \mathrm{Tr}[\mathcal{K}\mathcal{Q}\mathcal{H}] \tag{64}$$

where $\mathcal{Q} : L^2(\mathcal{U}) \to L^2(\mathcal{U})$ is the covariance operator associated with the Gaussian process covariance function $q$. This quantity is finite, since all of the operators involved are Hilbert-Schmidt and trace-class.

We now would like to show that the finite-dimensional notion of "decoding similarity" introduced in the main text (appropriately normalized) can be thought of as a plug-in estimator of the quantity eq. (64), and that this estimator converges to $\mathbb{E}_\eta[S]$ as $M \to \infty$.

A 'plug-in' estimator for this could be to use the Gram matrices for each kernel and matrix multiplication to approximate the integrals. First, we sample $M$ examples from the input distribution $\boldsymbol{u} \sim P_{\boldsymbol{u}}$ and construct the Gram matrices $\boldsymbol{Q}_{ij} = q(\boldsymbol{u}_i, \boldsymbol{u}_j)$ and

$$\boldsymbol{K}_{ij} = k(\boldsymbol{u}_i, \boldsymbol{u}_j) = f(\boldsymbol{u}_i)^\top \boldsymbol{G}(f)^{-1} f(\boldsymbol{u}_j) \tag{65}$$

$$\boldsymbol{H}_{ij} = h(\boldsymbol{u}_i, \boldsymbol{u}_j) = g(\boldsymbol{u}_i)^\top \boldsymbol{G}(g)^{-1} g(\boldsymbol{u}_j). \tag{66}$$

Then we have, by the strong law of large numbers,

$$\int_{\mathcal{U}} \int_{\mathcal{U}} \int_{\mathcal{U}} k(\boldsymbol{u}, \boldsymbol{u}') q(\boldsymbol{u}', \boldsymbol{u}'') h(\boldsymbol{u}, \boldsymbol{u}'')\, p(\boldsymbol{u}) d\boldsymbol{u}\, p(\boldsymbol{u}') d\boldsymbol{u}'\, p(\boldsymbol{u}'') d\boldsymbol{u}'' = \lim_{M \to \infty} \frac{1}{M^3} \sum_{ijk} K_{ij} Q_{jk} H_{ki}$$

or

$$\mathrm{Tr}[\mathcal{K}\mathcal{Q}\mathcal{H}] = \lim_{M \to \infty} \frac{1}{M^3} \mathrm{Tr}\,\boldsymbol{K}\boldsymbol{Q}\boldsymbol{H}. \tag{67}$$

However, the 'true' Gram matrices generated from the kernels in eq. (65) and eq. (66) in practice are also estimated, as the $N \times N$ regularization matrices $\boldsymbol{G}(f)$ and $\boldsymbol{G}(g)$ are potentially estimated from the same $M$ samples from $P_{\boldsymbol{u}}$. We estimate $\boldsymbol{G}(f)$ and $\boldsymbol{G}(g)$ using the plug-in estimates of the $N \times N$ covariance matrices $\mathbb{E}[f(\boldsymbol{u})f(\boldsymbol{u})^\top]$ and $\mathbb{E}[g(\boldsymbol{u})g(\boldsymbol{u})^\top]$.

$$\boldsymbol{G}(\boldsymbol{X}) = \frac{\alpha}{M} \boldsymbol{X}^\top \boldsymbol{X} + \beta \boldsymbol{I} \xrightarrow[M \to \infty]{} \alpha \mathbb{E}[f(\boldsymbol{u})f(\boldsymbol{u})^\top] + \beta \boldsymbol{I} = \boldsymbol{G}(f) \tag{68}$$

$$\boldsymbol{G}(\boldsymbol{Y}) = \frac{\alpha}{M} \boldsymbol{Y}^\top \boldsymbol{Y} + \beta \boldsymbol{I} \xrightarrow[M \to \infty]{} \alpha \mathbb{E}[g(\boldsymbol{u})g(\boldsymbol{u})^\top] + \beta \boldsymbol{I} = \boldsymbol{G}(g) \tag{69}$$

More precisely, it can be shown that the empirical regularization matrices $\boldsymbol{G}(\boldsymbol{X})$ and $\boldsymbol{G}(\boldsymbol{Y})$ concentrates around $\boldsymbol{G}(f)$ and $\boldsymbol{G}(g)$ in operator norm:

$$\|\boldsymbol{G}(\boldsymbol{X}) - \boldsymbol{G}(f)\|_{op} \xrightarrow[M \to \infty]{} 0 \tag{70}$$

$$\|\boldsymbol{G}(\boldsymbol{Y}) - \boldsymbol{G}(g)\|_{op} \xrightarrow[M \to \infty]{} 0 \tag{71}$$

Bounds on the rate of this convergence in operator norm can be obtained using elementary matrix concentration inequalities. We now must argue that $\boldsymbol{G}(\boldsymbol{X})^{-1}$ concentrates to $\boldsymbol{G}(f)^{-1}$ and similarly $\boldsymbol{G}(\boldsymbol{Y})^{-1}$ concentrates to $\boldsymbol{G}(g)^{-1}$. Since the matrices $\boldsymbol{G}(\boldsymbol{X})$ and $\boldsymbol{G}(f)$ are symmetric and positive definite, we have

$$\|\boldsymbol{G}(\boldsymbol{X})^{-1}\|_{op} = \max_{v:\|v\| \le 1} \|(\frac{\alpha}{M} \boldsymbol{X}^\top \boldsymbol{X} + \beta \boldsymbol{I})^{-1} v\|_2 = \frac{1}{\lambda_{min}(\boldsymbol{G}(\boldsymbol{X}))} \le \frac{1}{\beta} \tag{72}$$

$$\|\boldsymbol{G}(f)^{-1}\|_{op} = \max_{v:\|v\| \le 1} \|(\alpha \mathbb{E}[f(\boldsymbol{u})f(\boldsymbol{u})^\top] + \beta \boldsymbol{I})^{-1} v\|_2 = \frac{1}{\lambda_{min}(\boldsymbol{G}(f))} \le \frac{1}{\beta} \tag{73}$$

where the last inequality on the right hand side is a consequence of the Courant-Fischer theorem and the positive-semidefiniteness of the covariance matrix. Similar relations hold for $\boldsymbol{G}(\boldsymbol{Y})$ and $\boldsymbol{G}(g)$. Now we would like to study the quantity

$$\|(\boldsymbol{G}(\boldsymbol{X})^{-1} - \boldsymbol{G}(f)^{-1})\|_{op} = \|[(\frac{\alpha}{M} \boldsymbol{X}^\top \boldsymbol{X} + \beta \boldsymbol{I})^{-1} - (\alpha \mathbb{E}[f(\boldsymbol{u})f(\boldsymbol{u})^\top] + \beta \boldsymbol{I})^{-1}]\|_{op} \tag{74}$$

Multiplying the quantity inside the norm by identity and invoking the bound in eq. (72), we have

$$\|(\boldsymbol{G}(\boldsymbol{X})^{-1} - \boldsymbol{G}(f)^{-1})\|_{op} \le \frac{1}{\beta} \|\boldsymbol{I} - (\frac{\alpha}{M} \boldsymbol{X}^\top \boldsymbol{X} + \beta \boldsymbol{I})(\alpha \mathbb{E}[f(\boldsymbol{u})f(\boldsymbol{u})^\top] + \beta \boldsymbol{I})^{-1}\|_{op} \tag{75}$$

Pulling a factor of $(\alpha \mathbb{E}[f(\boldsymbol{u})f(\boldsymbol{u})^\top] + \beta \boldsymbol{I})^{-1}$ out of everything in the norm on the right hand side:

$$\|(\boldsymbol{G}(\boldsymbol{X})^{-1} - \boldsymbol{G}(f)^{-1})\|_{op} \le \frac{1}{\beta} \|[(\alpha \mathbb{E}[f(\boldsymbol{u})f(\boldsymbol{u})^\top] + \beta \boldsymbol{I}) \\ - (\frac{\alpha}{M} \boldsymbol{X}^\top \boldsymbol{X} + \beta \boldsymbol{I})](\alpha \mathbb{E}[f(\boldsymbol{u})f(\boldsymbol{u})^\top] + \beta \boldsymbol{I})^{-1}\|_{op}$$

and using the submultiplicative property of the operator norm,

$$\|(\boldsymbol{G}(\boldsymbol{X})^{-1} - \boldsymbol{G}(f)^{-1})\|_{op} \le \frac{|\alpha|}{\beta} \|\mathbb{E}[f(\boldsymbol{u})f(\boldsymbol{u})^\top] - \frac{1}{M} \boldsymbol{X}^\top \boldsymbol{X}\|_{op} \|(\alpha \mathbb{E}[f(\boldsymbol{u})f(\boldsymbol{u})^\top] + \beta \boldsymbol{I})^{-1}\|_{op}.$$

Using eq. (72) again, we have

$$\|(\boldsymbol{G}(\boldsymbol{X})^{-1} - \boldsymbol{G}(f)^{-1})\|_{op} \le \frac{|\alpha|}{\beta^2} \|\mathbb{E}[f(\boldsymbol{u})f(\boldsymbol{u})^\top] - \frac{1}{M} \boldsymbol{X}^\top \boldsymbol{X}\|_{op} \tag{76}$$

or

$$\|(\boldsymbol{G}(\boldsymbol{X})^{-1} - \boldsymbol{G}(f)^{-1})\|_{op} \le \frac{|\alpha|}{\beta^2} \|\boldsymbol{G}(\boldsymbol{X}) - \boldsymbol{G}(f)\|_{op} \tag{77}$$

with an analogous bound holding for $\boldsymbol{G}(\boldsymbol{Y})$ and $\boldsymbol{G}(g)$.

With this knowledge, we define $\hat{\boldsymbol{K}}$ and $\hat{\boldsymbol{H}}$ as the empirical Gram matrices constructed using the estimates $\boldsymbol{G}(\boldsymbol{X})$ and $\boldsymbol{G}(\boldsymbol{Y})$, and conclude that, for every $\{i, j\}$,

$$\hat{\boldsymbol{K}}_{ij} = f(\boldsymbol{u}_i)^\top \boldsymbol{G}(\boldsymbol{X})^{-1} f(\boldsymbol{u}_j) \xrightarrow[M \to \infty]{} f(\boldsymbol{u}_i)^\top \boldsymbol{G}(f)^{-1} f(\boldsymbol{u}_j) = \boldsymbol{K}_{ij} \tag{78}$$

$$\hat{\boldsymbol{H}}_{ij} = g(\boldsymbol{u}_i)^\top \boldsymbol{G}(\boldsymbol{Y})^{-1} g(\boldsymbol{u}_j)^\top \xrightarrow[M\to\infty]{} g(\boldsymbol{u}_i)^\top \boldsymbol{G}(g)^{-1} g(\boldsymbol{u}_j) = \boldsymbol{H}_{ij}. \tag{79}$$

where we recognize the matrices $\hat{\boldsymbol{K}}$ and $\hat{\boldsymbol{H}}$ as $M\boldsymbol{K}_X$ and $M\boldsymbol{K}_Y$ in the main text.

Lastly, we would now like to study

$$\left| \frac{1}{M^3} \operatorname{Tr} \hat{\boldsymbol{K}} \boldsymbol{Q} \hat{\boldsymbol{H}} - \operatorname{Tr} \mathcal{K}\mathcal{Q}\mathcal{H} \right| \tag{80}$$

taking note that the quantity $\frac{1}{M^3} \operatorname{Tr} \hat{\boldsymbol{K}} \boldsymbol{Q} \hat{\boldsymbol{H}}$ is a scaled version of eq. (13) in the main text. The triangle inequality implies a relationship with the trace of the true Gram matrices:

$$\left| \frac{1}{M^3} \operatorname{Tr} \hat{\boldsymbol{K}} \boldsymbol{Q} \hat{\boldsymbol{H}} - \operatorname{Tr} \mathcal{K}\mathcal{Q}\mathcal{H} \right| \leq \left| \frac{1}{M^3} \operatorname{Tr} \hat{\boldsymbol{K}} \boldsymbol{Q} \hat{\boldsymbol{H}} - \frac{1}{M^3} \operatorname{Tr} \boldsymbol{K} \boldsymbol{Q} \boldsymbol{H} \right| + \left| \frac{1}{M^3} \operatorname{Tr} \boldsymbol{K} \boldsymbol{Q} \boldsymbol{H} - \operatorname{Tr} \mathcal{K}\mathcal{Q}\mathcal{H} \right|. \tag{81}$$

The first term on the right hand side can be bounded:

$$\begin{aligned}
\left| \frac{1}{M^3} \operatorname{Tr} \hat{\boldsymbol{K}} \boldsymbol{Q} \hat{\boldsymbol{H}} - \frac{1}{M^3} \operatorname{Tr} \boldsymbol{K} \boldsymbol{Q} \boldsymbol{H} \right| &\leq \frac{1}{M^3} \sum_{ijk} \left| \hat{\boldsymbol{K}}_{ij} \boldsymbol{Q}_{jk} \hat{\boldsymbol{H}}_{ki} - \boldsymbol{K}_{ij} \boldsymbol{Q}_{jk} \boldsymbol{H}_{ki} \right| \\
&= \frac{1}{M^3} \sum_{ijk} \left| \boldsymbol{Q}_{jk} \right| \left| \hat{\boldsymbol{K}}_{ij} \hat{\boldsymbol{H}}_{ki} - \boldsymbol{K}_{ij} \boldsymbol{H}_{ki} \right| \\
&= \frac{1}{M^3} \sum_{ijk} \left| \boldsymbol{Q}_{jk} \right| \left| (\hat{\boldsymbol{K}}_{ij} - \boldsymbol{K}_{ij}) \hat{\boldsymbol{H}}_{ki} + \boldsymbol{K}_{ij} (\hat{\boldsymbol{H}}_{ki} - \boldsymbol{H}_{ki}) \right| \\
&= \frac{1}{M^3} \sum_{ijk} \left| \boldsymbol{Q}_{jk} \right| \left( \left| \hat{\boldsymbol{K}}_{ij} - \boldsymbol{K}_{ij} \right| \left| \hat{\boldsymbol{H}}_{ki} \right| + \left| \boldsymbol{K}_{ij} \right| \left| \hat{\boldsymbol{H}}_{ki} - \boldsymbol{H}_{ki} \right| \right).
\end{aligned}$$

Using the definitions of $\hat{\boldsymbol{K}}_{ij}$, $\boldsymbol{K}_{ij}$, $\boldsymbol{H}_{ij}$, and $\hat{\boldsymbol{H}}_{ij}$,

$$\begin{aligned}
\left| \frac{1}{M^3} \operatorname{Tr} \hat{\boldsymbol{K}} \boldsymbol{Q} \hat{\boldsymbol{H}} - \frac{1}{M^3} \operatorname{Tr} \boldsymbol{K} \boldsymbol{Q} \boldsymbol{H} \right| &\leq \frac{1}{M^3} \| (\boldsymbol{G}(\boldsymbol{X})^{-1} - \boldsymbol{G}(f)^{-1}) \|_{op} \sum_{ijk} \left| \boldsymbol{Q}_{jk} \right| \left| \hat{\boldsymbol{H}}_{ki} \right| \| f(\boldsymbol{u}_i) \| \| f(\boldsymbol{u}_j) \| \\
&\quad + \frac{1}{M^3} \| (\boldsymbol{G}(\boldsymbol{Y})^{-1} - \boldsymbol{G}(g)^{-1}) \|_{op} \sum_{ijk} \left| \boldsymbol{Q}_{jk} \right| \left| \boldsymbol{K}_{ij} \right| \| g(\boldsymbol{u}_i) \| \| g(\boldsymbol{u}_j) \|.
\end{aligned}$$

Since $\left| \boldsymbol{Q}_{jk} \right|$, $\left| \hat{\boldsymbol{H}}_{ki} \right|$, and $\left| \boldsymbol{K}_{ij} \right|$ are assumed to be finite for all $i, j, k \in \{1, ..., M\}$, and the norms of the neural responses $\| f(\boldsymbol{u}_i) \|$ and $\| g(\boldsymbol{u}_i) \|$ can be assumed to be bounded[4] for all $i \in \{1, ..., M\}$, we can conclude by referencing eq. (77) that the right hand side approaches 0 as $M \to \infty$. This argument can be made precise by considering the rate of convergence of $\hat{\boldsymbol{K}}_{ij} \to \boldsymbol{K}_{ij}$ and $\hat{\boldsymbol{H}}_{ij} \to \boldsymbol{H}_{ij}$ that is inherited from the concentration of $\boldsymbol{G}(\boldsymbol{X}) \to \boldsymbol{G}(f)$ and $\boldsymbol{G}(\boldsymbol{Y}) \to \boldsymbol{G}(g)$.

The second term on the right hand side of eq. (81) is precisely the convergence of a sum over 'true' Gram matrices to the trace of a composition of integral operators described in eq. (67), so this term also approaches 0 as $M \to \infty$.

Putting it all together

$$\frac{1}{M^3} \operatorname{Tr} \hat{\boldsymbol{K}} \boldsymbol{Q} \hat{\boldsymbol{H}} \xrightarrow[M\to\infty]{} \operatorname{Tr}[\mathcal{K}\mathcal{Q}\mathcal{H}] = \mathbb{E}_\eta[S]. \tag{82}$$

---

[4]Similar to the treatment in [28], this could be interpreted as a resource constraint on the neural responses.

where we can recognize the plug-in estimate as

$$\frac{1}{M^3} \operatorname{Tr} \hat{\boldsymbol{K}} \boldsymbol{Q} \hat{\boldsymbol{H}} = \frac{1}{M} \operatorname{Tr} \boldsymbol{K}_X \boldsymbol{K}_z \boldsymbol{K}_Y = \frac{1}{M} \mathbb{E} \langle \boldsymbol{X} \boldsymbol{w}^*, \boldsymbol{Y} \boldsymbol{v}^* \rangle. \tag{83}$$

in the notation used in the main text (compare with eq. (13)).

