# OpenReview forum: "What Representational Similarity Measures Imply about Decodable Information"
_NeurIPS.cc/2024/Workshop/UniReps — UniReps_

### Official Review · Reviewer_Aq6y · 2024-10-05
**Interesting paper that aims to link geometric and decoding perspectives in similarity measures**

**Rating:** 7
**Confidence:** 4

**Review:**

The paper studies how representational similarity measures like Centered Kernel Alignment (CKA), Canonical Correlation Analysis (CCA), and Procrustes Shape Distance can be understood through decodable information. Specifically, the authors propose a framework that links these similarity measures to the ability of neural representations to linearly decode information across a distribution of tasks. The paper formalizes these connections by analyzing the alignment of linear decoders and deriving bounds for different similarity measures.

Strengths:
- The findings on the relationship between similarity measures and linear decoders have the potential to impact a range of fields, including neuroscience and representation learning.
- Solid theory: The authors present detailed mathematical proofs and derive bounds that formalize the relationships between different similarity measures.

Weaknesses:
- The framework is built on the assumption that the distribution of decoding targets satisfies $E[zz^{\top}]=I$, but this condition may not be met in practical settings.
- The paper’s framework is heavily focused on linear decoders.
- Despite the strong theoretical foundations, the paper does not provide empirical evidence to show how the proposed measures perform on actual neural networks or datasets.

---

### Official Review · Reviewer_JAo1 · 2024-10-06
**Interesting framework for unifying existing geometric measures - lacking presentation**

**Rating:** 6
**Confidence:** 4

**Review:**

This work puts forward a framework to unify different representational similarity measures (e.g., CKA, CCA, Procrustes distance) via the concept of linear decodability. The authors show that several measures can be interpreted as “alignment scores” between optimal linear decoders, thus connecting geometric similarity with decoding similarity. This is a key contribution that appears novel and important to me.

However, it must be said that some of the proven connections are not particularly surprising: linear similarity and linear decodability can be expected to be closely related. I also think that the presentation could be significantly improved as the paper is rather hard to read and appreciate. On this, I would have three specific suggestions.

1) The main findings of the paper are hard to quickly find and grasp. In this respect, a summary table or summary figure (e.g., outlining the correspondences between the different measures) would **drastically** improve the readability of the paper.
2) Related to 1. I would personally appreciate a presentation focusing on the results and building an intuition around them, with proofs either completely moved to the Appendix or sketched in the main text but isolated in “skippable” sections.
3) Finally, I think the paper would benefit from a discussion of the potential practical implications of the framework, which instead appears completely missing from the current version.

All things considered, I think this is a potentially a clear “accept” (7) in terms of the results, but the presentation is so lacking to significantly lower my rating.

---

### Official Review · Reviewer_QGzs · 2024-10-06
**Theoretically well motivated and justified framework but would be more convincing with empirical evaluation.**

**Rating:** 8
**Confidence:** 3

**Review:**

## Summary of the paper

The paper proposes a novel framework to compare representations from different models on the basis of the functions that they implement — functions that can be decoded from them. They argue that representation geometry should be considered a proxy for model-implemented functions, and comparisons of representation geometries should be done with this in mind.

They formalize a framework for representational similarity defined based on the algorithmic expressivity of the representations. As a stand-in for the functions implementable by the models, they assume the Linear Representation Hypothesis i.e.  they characterize the representation power with respect to a function as the ability to linearly simulate or “extract” that task from the representations — a practice that is popular in probing for interpretability literature also.

This allows them to define a notion of the “most decodable” and “least decodable” functions i.e. functions that are the easiest and the hardest to extract from the representations.

Under specific assumptions, the authors demonstrate that their framework becomes equivalent to popular second-order similarity metrics, such as Centered Kernel Alignment (CKA).

## Strengths

- The proposed framework is well motivated — it tries to characterize the relationship of representation geometry to expressivity of representations.
- The framework provides meaningful connections with existing popular and well-motivated similarity metrics.

## Weakness

- There is no empirical validation of their framework. Often, a lot of nice seemingly well motivated properties together lead to unexpected and perhaps undesirable consequences. Therefore, I would have been more convinced if there was empirical demonstration that the metric assigns expected values.
- The meaning of the “least decodable” setting is unclear — why would this not be a function with pseudo-random outputs and if so, why is that a meaningful thing to look at?
- The E[zz^T] = I assumption could use qualification with respect to real data. It seems arbitrary but is important for the relationship to CKA.

## Experiment Suggestion

The metric should consider demonstrating their framework by measuring similarities across models while controlling for the expressive power (in terms of number of parameters or performance on relevant NLP benchmarks)

## Conclusion

While the paper presents a theoretically interesting and well-motivated framework, it would benefit significantly from empirical validation and clearer justification of its assumptions. The relationship between representation geometry and functional expressivity is a valuable contribution to the field, but practical demonstrations would strengthen the paper's impact.

---

### Official Review · Reviewer_frEY · 2024-10-07
**Clarifying and straightforward theoretical contribution**

**Rating:** 9
**Confidence:** 4

**Review:**

This was an nice paper to read, and makes a point that is easy to remember. The authors study a family of linear regression objectives subject to some kind of norm regularization, and show that some popular measures of representation alignment are related to the predictions of linear decoders trained on each representation. My comments are mostly tossing up things I might look forward to in a future instance of this work.

I would be interested in seeing how these results relate to similar ones established previously. In particular, I'm thinking of:

(1) the result from Cristianini et al. (2001) showing that the (uncentered) kernel alignment is related to generalization of a certain binary classifier. Cortes et al. (2012) also showed this (just empirically I think) for centered alignment.

(2) Gretton et al. (2005) introduced something like the CKA as a test of statistical independence. Specifically, they relate the (numerator of) the CKA to the norm of a "cross-covariance" operator between X and Y, and I think for the choice of a linear kernel (where the RKHS is linear decoders) that might imply some results similar to those of the paper?

I only bring these up in case their frameworks offer an avenue for the kinds of extensions mentioned in the discussion. They could also be good additions to a "related work" section of this paper, but I don't feel strongly about that.

On a more substantive side, it seems like a focus of the paper is on unifying these measures, but I was interested if this framework can help to explain some of their observed discrepancies. I'm thinking, for example, of the recent work by Soni et al. (2024), showing that choice of metric can have a pretty big difference on alignment scores. If geometric alignment is related to functional alignment, it seems important to know why different measures of alignment produce different results. Might be a nice application of this work.

At the moment the numerics are a fairly small part of the paper. A nice proof of principle could be to apply this to artificial networks trained on (simulated) tasks of varying similarity, and to see how much the geometric similarity actually reflects functional similarity, and whether this framework is helpful in understanding those results.

links:

Cristianini et al. (2001): https://papers.nips.cc/paper_files/paper/2001/hash/1f71e393b3809197ed66df836fe833e5-Abstract.html

Gretton et al. (2005): https://www.gatsby.ucl.ac.uk/~gretton/papers/GreBouSmoSch05.pdf

Cortes et al. (2012): https://www.jmlr.org/papers/volume13/cortes12a/cortes12a.pdf

Soni et al. (2024): https://www.biorxiv.org/content/10.1101/2024.08.07.607035v1

---

### Decision · Program_Chairs · 2024-10-10

**Decision:**

Accept (Oral)

**Comment:**

In light of the reviewers' feedback and relevancy of the submission, we are pleased to accept this paper for presentation at UniReps 2024. We kindly ask the authors to incorporate the reviewers' suggestions and feedback in the final camera-ready version of the manuscript.